# The Functional Morphology of the Bursa Copulatrix of a Butterfly That Does Not Digest Spermatophores (*Leptophobia aripa*, Pieridae)

**DOI:** 10.3390/insects13080714

**Published:** 2022-08-08

**Authors:** David Xochipiltecatl, Carlos Cordero, Joaquín Baixeras

**Affiliations:** 1Posgrado en Ciencias Biológicas, Instituto de Ecología, Universidad Nacional Autónoma de México, Ciudad de Mexico 04510, Mexico; 2Departamento de Ecología Evolutiva, Instituto de Ecología, Universidad Nacional Autónoma de México, Ciudad de Mexico 04510, Mexico; 3Institut Cavanilles de Biodiversitat i Biologia Evolutiva, Universitat de Valencia, Paterna, 6980 Valencia, Spain

**Keywords:** Lepidoptera, Pieridae, female genitalia, corpus bursae, ductus bursae, spermatophore, sperm transfer

## Abstract

**Simple Summary:**

Male butterflies transfer sperm to the female within a package of secretions named spermatophore. These secretions include nutritious substances (butterfly spermatophores are considered nuptial gifts) and compounds that influence different reproductive processes of females in a hormone-like way. During copulation, the spermatophore is deposited in a complex female bag-like organ known as the bursa copulatrix, where it is processed and digested. Thus, the bursa copulatrix mediates male–female interactions during and after copulation that are crucial to the reproductive success of males and females. We studied *Leptophobia aripa*, a common butterfly in Central Mexico that, contrary to what has been observed in most butterfly species previously studied, does not digest the spermatophore (i.e., spermatophores are not nutritious nuptial gifts in this species). We present a detailed description of the different elements of the bursa copulatrix and of its interaction with the spermatophore. We provide a functional interpretation of these interactions and propose a novel mechanism for the transfer of the sperm from the spermatophore to its final storage organ (another bag-like organ named the spermatheca).

**Abstract:**

The bursa copulatrix of female Lepidoptera is a complex organ where crucial male–female reproductive interactions occur during and after copulation. The bursa copulatrix receives, stores, and digests the spermatophore and other substances transferred by the male during copulation, and is involved in changes in female receptivity, ovogenesis, and oviposition. Although females of the butterfly *Leptophobia aripa* do not digest the spermatophore, they possess a prominent signum. Since, in general, the function of the signum is considered to be the piercing or tearing of the spermatophore to initiate its digestion, its presence in *L. aripa* poses a conundrum. We undertook a microscopic study of the different components of the bursa copulatrix (ductus bursae and corpus bursae) and found structural differences that we interpreted in functional terms. We provide a detailed description of the signum and present experimental data regarding its effect on the spermatophore. Our observations led us to propose a novel hypothesis regarding the function of the signum.

## 1. Introduction

The bursa copulatrix of female Lepidoptera is a genital organ where crucial male-female reproductive interactions occur during and after copulation [1,2]. The main body of the bursa copulatrix, the corpus bursae (CB hereafter), is a complex bag-like organ (Figure 1A) that performs multiple functions, including receiving, storing, and digesting the spermatophore and other substances transferred by the male during copulation; the CB is also involved in the control of the sexual receptivity and oviposition behavior of the mated female [1,3,4,5,6,7]. Generally, the CB connects with the genital pore, the ostium, through a tubular duct called the ductus bursae that can be straight or convoluted (Figure 1A). The area of the CB where it meets the ductus bursae is called the cervix bursae. It may be a rather simple transitional funnel-like area but quite often presents sclerotizations. In some cases, a duct is not discernible and practically the bursa copulatrix opens directly to the exterior through the ostium. In many species, the inner wall of the CB includes sclerotizations, inward-protruding structures, collectively known as signa (Figure 1A,B), which may bear sharp edges and/or acute points [1,5,8,9,10]. Although there are several hypotheses on the function of signa, most available evidence suggests that these structures are used to break the spermatophore envelope [5,8,11]. After the spermatophore is deposited, the muscles enveloping the CB, some of which are inserted in the base of the signum, start contracting [3,4], helping the signa to pierce and tear the spermatophore envelope [4,5,11,12], thus facilitating the mechanical digestion of the spermatophore [5,6,7].

We recently reported that females of the butterfly *Leptophobia aripa* (Boisduval, 1836) (Pieridae) do not digest the spermatophore [2]. Since we also found that their CB lacks a muscular sheath and pores on its inner epithelium, we proposed that female *L. aripa* lacks the “apparatus” required for the mechanical digestion of the spermatophore and for the absorption of its contents [2]. Thus, the presence of a strongly sclerotized signum by the cervix, and associated muscles, seems paradoxical. Here, we provide details of the structure of the bursa copulatrix of *L. aripa* and infer its functional role. We describe the unusual signum, its muscular assemblage, and its interaction with the spermatophore. We also propose a novel functional hypothesis regarding the signum.

## 2. Materials and Methods

In this paper, we present original, unpublished observations and results, mostly obtained from the examination of the same specimens undertaken in a previous study [2]. We studied the bursa copulatrix of females processed in three different ways:

(a)Forty-eight mated females raised in the laboratory from eggs laid by females collected in the Ciudad Universitaria campus of the Universidad Nacional Autónoma de México (CU-UNAM) in México City. The wild females were kept in one-liter plastic containers, fed a 10% sugar solution daily, and allowed to lay eggs on fresh leaves of *Tropaeolum major* (Tropaeolaceae). Larvae were individually reared on *T. major* leaves within plastic containers. Mated females were obtained by placing unmated males and females in cylindrical mesh cloth cages (25 cm diameter × 60 cm height) in a garden located at CU-UNAM. Mated females were sacrificed by freezing (−70 °C) at different times after copulation, from a few minutes to 96 h.(b)Six unmated females raised in the laboratory as explained above. Four of these females were placed in a freezer (−20 °C) for a few minutes and then injected with Karnovsky fixative in the body cavity; the other two females were sacrificed by freezing and kept in glassine envelops until dissection.(c)Three mated females collected in the field while lying eggs. These females were allowed to lay eggs for a few days in the laboratory; then, they were sacrificed by freezing (−70 °C) and their abdomens were preserved in 100% ethylic alcohol.

Frozen females were thawed at ambient temperature and their abdomens were opened and cleaned with fine forceps. The bursa copulatrix and spermatophores were dissected, and careful observations were made and photographs were obtained. All specimens were dissected under stereomicroscopes (Leica^TM^ MZ8 and Olympus^TM^ SZH10). Optical microscope photographs were obtained with a Leica^TM^ Z16 equipment and the Z-stacks technique was thoroughly used. Observations and microphotographs were also made with a scanning electron microscope (Hitachi^TM^ S4800). The samples for SEM were prepared following standard methods [2,10]. Briefly, the clean CB and DB were stained with 2% osmium tetroxide for 20 min, washed with water, placed in microporous specimen capsules, and dehydrated in increased grade ethanol. The CB and DB were dried to critical point in an Autosamdry 814 (Tousimis^TM^), placed on SEM stubs using carbon tape and silver conducting paint and sputtered with Au-Pd. Further details can be found in [2].

Genitalia terminology follows Klots [9]. Terminology of the spermatophore follows Mann [12], except for the spermatophore “corpus” which, in agreement with Drummond [13] and Hague et al. [14], we call the “bulb” to avoid confusion with the “corpus bursae”.

## 3. Results

The internal genitalia of the female *L. aripa* extends along approximately half of the distended abdomen (Figure 1A). This abdomen has extensive membranous pleural areas (Figure 1A). The ostium of the genitalia of *L. aripa* opens in the depth of a broad funnel-like bilobed sterigma (Figure 1A). The membranous ductus bursae (DB hereafter) is attached to the sterigma through a sclerotized ring to which the ductus seminalis (the duct through which sperm leaves the bursa copulatrix and travels in the direction of the spermatheca) also connects (Figure 1A). The DB is somewhat dilated in front of this ring. The anterior part of the DB is longitudinally folded and internally covered by aligned fields of acanthae (Figure 2A). The CB is a relatively simple spherical bag distended in the left ventrolateral area (Figure 1B). The integument is membranous, and the internal surface appears unornamented, lacking acanthae or any kind of micro-teeth (Figure 2B). The differences in ornamentation between the internal surfaces of the DB and the CB are clearly shown in Figure 2C.

The signum, arranged ventrolaterally left, consists of a transversely elongate plate embracing approximately 50% of the circumference of the cervix. The central part of this plate is narrowed and indented, and protrudes inward (Figure 2C). Muscle fibers attach to the lateral and central areas of the signum and run longitudinally along the DB (Figure 2D). No muscle layer is detected anterior to the cervix (Figure 2D), i.e., on the CB [2].

A single spermatophore was found in the bursa copulatrix of all mated females (Figure 3A). The main body of the spermatophore (the bulb) is anteriorly bilobed and practically fills the whole lumen of the CB (Figure 3B), although some other substances are usually observable inside the CB. The bulb extends into a hollow tube (the collum) inside the DB (Figure 3A) to the level of the junction of the ductus seminalis, where it is plugged by an irregular mass of denser material (frenum). The main axis of the bulb and the collum may be aligned or bent at an angle of approximately 45° just at the level of the signum (Figure 3A). The surface of the bulb appears smooth and intact (Figure 2E) just as the internal side of the CB. In contrast, the surface of the collum shows the imprint of the acanthae lining of the DB (Figure 2E). A clearly noticeable indented area, a complementary negative mark of the signum, is present at the level of the base of the collum, in the transition to the bulb (Figure 2E, see also Figure 3). The imprint of the signum was present even in the spermatophores contained in females killed immediately after the end of copulation. No portion of the spermatophore appears perforated or broken. Under the optical microscope, the bulb shows a thin but resistant wall. The interior of the bulb appears to be filled with granular material, especially at the anterior part, which becomes progressively more transparent with denser granules towards the collum (Figure 3B). The scanning electron microscope revealed a thin but relatively hard wall and a rather spongy material filling most of the inside of the spermatophore (Figure 4A), except for a posterior cavity connected to the duct running along the collum (the “sperm sac” [4,12]) (Figure 4B).

## 4. Discussion

Our detailed observations of the spermatophore, bursa copulatrix, and associated muscles led us to the following functional interpretations. The precise positioning of the tip of the collum, where its opening is usually present, beside the junction with the ductus seminalis, and its fixation with a (presumably) male-secreted mass (the frenum), indicates the importance of the alignment of the opening of the collum and the opening of the ductus seminalis for the successful transfer of the spermatozoids from the spermatophore to the spermatheca. Further evidence in support of this interpretation is the imprint of the acanthae on the surface of the spermatophore collum, indicating that the DB firmly holds the spermatophore by the collum preventing it from displacing or turning on itself (Figure 2E). It is interesting to consider the presence of this attachment device (in the sense of Gorb [15]) in the DB of a species whose CB cannot firmly hold the spermatophore because it lacks a muscle sheath and internal micro-protuberances [2].

The muscles attached between the DB and signum when contracting are responsible for bending the spermatophore. The wide space of the abdominal cavity does not seem to be an obstacle for this. When the cervix of the CB is bent, the signum is in close contact with the collum, protruding into the central part in such a way that the lumen of the collum is blocked (Figure 2E). The observation of the imprint of the signum on the spermatophore in females sacrificed a few minutes after finishing mating indicates that the interaction between the signum and the spermatophore occurs before the end of copulation (once the spermatophore is fully formed but before separation of the couple) or very soon after separation of the couple. However, the deformation is detectable even in spermatophores removed from females killed 96 h after copulation. Whether this flexion of the bursa copulatrix occurs only once or repeatedly is uncertain. However, the existence of a certain variation in the angle of flexion of the spermatophores would be more compatible with the existence of repeated contractions.

The observations of the internal structure of the spermatophore revealed a cavity in the bulb connected to the collum (Figure 4B). According to Mann [12], this cavity is the sperm sac and contains the spermatozoids. The position of the sperm sac in *L. aripa* is almost identical to that observed in *Pieris brassicae* [4]. The mechanisms by which spermatozoa are evacuated from the spermatophore in Lepidoptera have not yet been fully elucidated, but passive transport from within the spermatophore is undoubtedly a critical factor [4,12,14]. The case of *L. aripa*, in which the spermatophore envelope remains intact after mating, is enigmatic. The spermatozoa have to reach the ductus seminalis along the hollow collum. However, the internal pressure in an undigested, slightly deformable spermatophore must be comparatively low. The existence of an intermediate valve within the spermatophore may well contribute to avoiding the recoil of the spermatozoa in the absence of optimal pressure. Subtle contractions and relaxations of the muscles of the cervix could help the signum to regulate the pressure between the collum and the sperm sac to facilitate the passage of the spermatozoa, minimizing the need for a higher pressure and subsequent liquid demand (see Figure 5 for a schematic representation of this hypothesis). Galicia et al. [5] reviewed the functions of the signa but, among them, they did not mention the possibility that the signum could act as a valve. This is the first time that this possibility has been mentioned.

Comparative evidence suggests that signa originally evolved to break spermatophore envelopes in polyandrous species, and that the evolution of monandry selects for the loss of signa [16]. However, in *L. aripa*, the evolution of monandry is associated with the evolutionary loss of the muscular sheath of the CB [2]. The lack of these muscles not only prevents the breaking of spermatophores, but also the ability to exert a pressure on the spermatophore, which may help the passage of sperm cells from the sperm sac to the spermatheca. This may explain why *L. aripa* females did not lose their signum and why the signum acquired a novel function that helps the transfer of sperm cells from the spermatophore to the spermatheca. It will be interesting to investigate if the signum of *L. aripa* has evolved modifications that improve its (possible) function as a valve, by comparison with the signa of closely related polyandrous species that digest the spermatophore.

## Figures and Tables

**Figure 1 insects-13-00714-f001:**
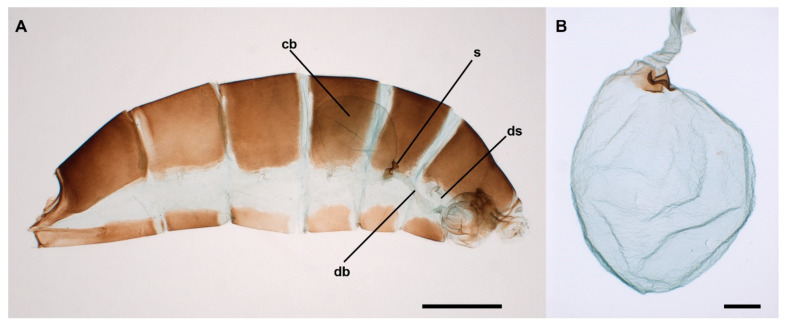
Bursa copulatrix of the butterfly *L. aripa*. (**A**) Cleared female abdomen showing the location and area occupied by the corpus bursae (cb), ductus bursae (db), ductus seminalis (ds), and signum (s). (**B**) Isolated CB of a virgin female. Figure 1B is a different photograph of the CB shown in Figure 1A in reference [2]. Scale bars: A = 1 mm, B = 200 µm.

**Figure 2 insects-13-00714-f002:**
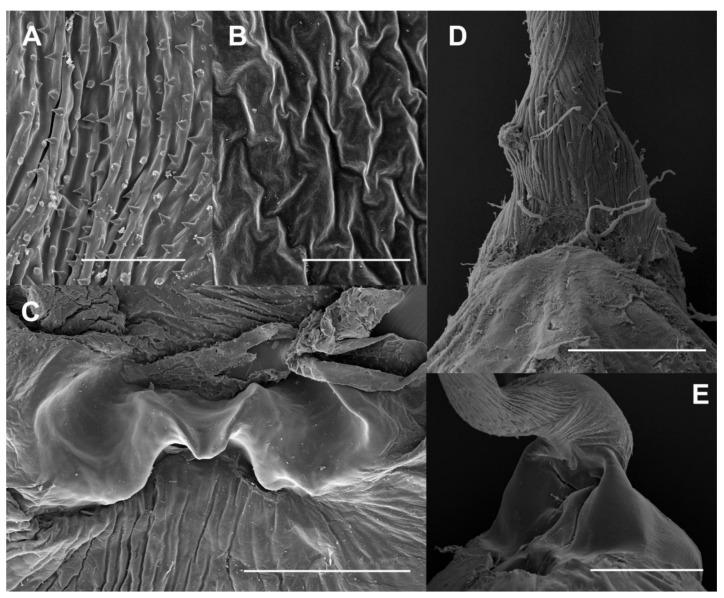
SEM images of the corpus bursae (CB) and spermatophore of the butterfly *L. aripa*. (**A**) Integument of the anterior part of the ductus bursae (DB) with an internal surface longitudinally folded and aligned fields of acanthae. (**B**) Integument of the CB showing the membranous and unornamented internal surface. (**C**) Integument of the CB cervix area, showing the signum formed by a transversely elongate plate with a narrowed and indented central part protruding inward. Note the differences in ornamentation between the internal surfaces of the DB (above the signum) and the CB (below the signum). (**D**) Close-up of the exterior part of the cervix (DB above, CB below), showing the muscle fibers attached to the lateral and central areas of the signum and fibers that run longitudinally along the DB. Note that no muscle layer is observed on the CB. (This photograph is the same as that in Figure 4E of reference [2]) (**E**) Close-up of the external surface of the spermatophore in the area that is in contact with the cervix of the CB (collum above, bulb below). Scale bars: A = 40 µm, B = 5 µm, C = 200 µm, D = 300 µm, E = 200 µm.

**Figure 3 insects-13-00714-f003:**
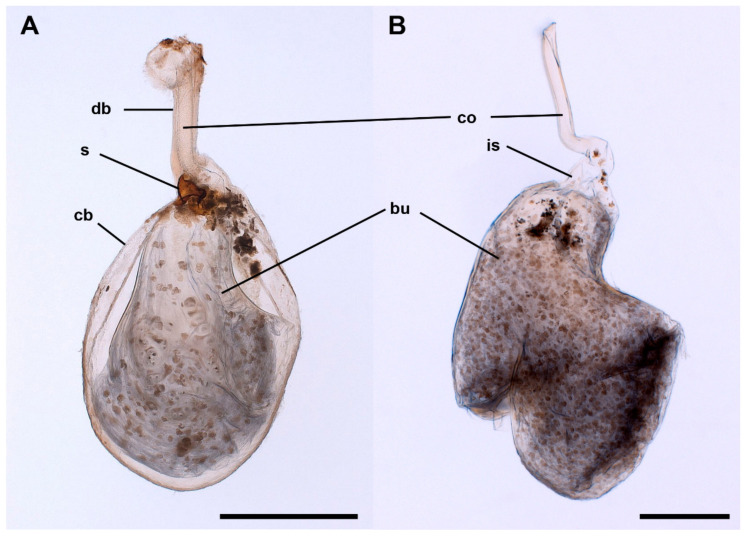
General shape of the spermatophore of *L. aripa*. (**A**) CB with spermatophore inside. The spermatophore fills the CB (cb) and extends into a hollow tube (collum) inside the DB (db). Note the bending of the main axis of the bulb (bu) and the collum (co) at an angle of approximately 45° at the level of the signum (s). (**B**) Isolated spermatophore showing that the bulb its anteriorly bilobed. Note the imprint of the signum (is) and the approximately 45° bend in the main axis of the bulb and the collum. Scale bars: A = 1 mm, B = 500 µm.

**Figure 4 insects-13-00714-f004:**
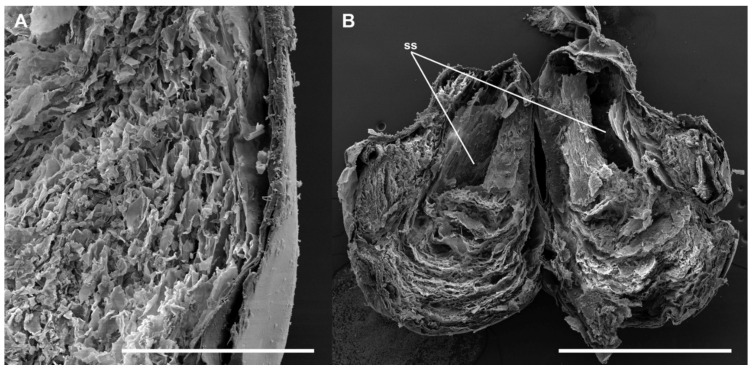
Inner structure of a spermatophore of *L. aripa* seen under the SEM. (**A**) Close-up of a portion of the wall of the spermatophore bulb showing a spongy material. (**B**) Inner structure of the spermatophore bulb and collum, showing the cavity in the posterior part of the bulb (called the sperm sac, ss) connected to the hollow duct and running along the collum. Scale bars: A = 200 µm, B = 1 mm.

**Figure 5 insects-13-00714-f005:**
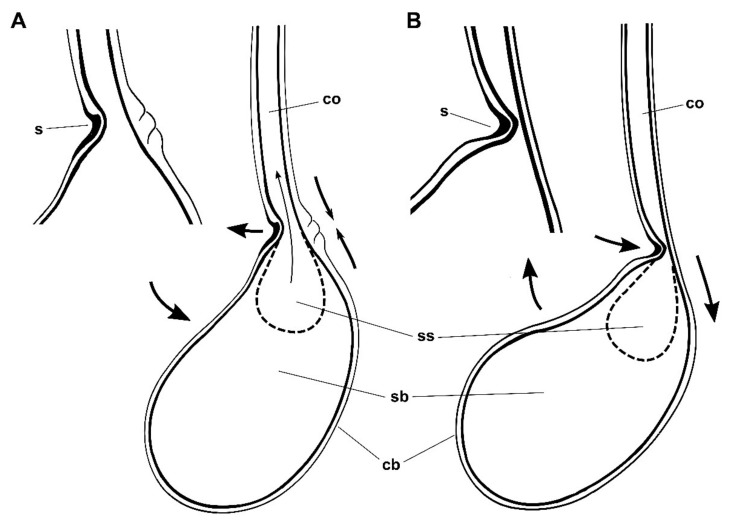
Hypothetical mechanism of the signum and associated muscles of female *L. aripa*. (**A**) Relaxed position. (**B**) Contracted position. When muscles contract, the cervix of the CB (cb) bends and the signum (s) presses the collum (co) with the protruding central part, and the deformation blocks the communication between the lumen of the collum and the sperm sac (ss). Note how this action bends the main axis of the bulb (sb) and the collum at an angle of approximately 45° and gives the base of the collum its curved shape.

## Data Availability

Data sharing is not applicable to this paper.

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
