# Peer review of "The Functional Morphology of the Bursa Copulatrix of a Butterfly That Does Not Digest Spermatophores (Leptophobia aripa, Pieridae)"

_insects, 2022, doi:10.3390/insects13080714_

Round 1
Reviewer 1 Report
This is a very fine contribution to the knowledge of the functional morphology of the bursa copulatrix of a peculiar butterfly species whose females do not digest the spermatophores.
Using careful, detailed observations, the authors provide an excellent explanation of the functioning of the structures in the studied species. Furthermore, they propose for the first time the possibility that the signum could act as a valve.
All the interpretations are supported by high quality illustrations.
I think this manuscript must be accepted in the current form. However I not assessed the English as I am not a native English speaker.
I only found that "B" is absent in the corresponding part of the Figure 1.
Congratulations to the authors for this excellent contribution!!!
Author Response
Dear Editor:
We thank reviewer 1 for his kind words.
In the revised version, we highlighted in yellow the deleted parts and in green the new parts.
We hope that you find the revised version of our manuscript acceptable for publication and look forward to hearing from you.
Best regards,
Carlos Cordero
Reviewer 2 Report
This is an interesting paper that I think ought to be published. The proposed mechanism of action is new and well documented
Author Response
Dear Editor:
We thank reviewer 2 for his kind words.
In the revised version, we highlighted in yellow the deleted parts and in green the new parts.
We hope that you find the revised version of our manuscript acceptable for publication and look forward to hearing from you.
Best regards,
Carlos Cordero
Reviewer 3 Report
Comment to the authors:
The authors investigated the structure of Bursa copulatrix in Leptophobia aripa (Pieridae), a butterfly considered as monandrous species, by using optical microscope and SEM. Contrary to most of butterfly species previously observed, the Bursa copulatrix of L. aripa does not digest the spermatophore that deposited from the male during copulation, but it does have the signa that are usually to piecer or tear the envelop of the spermatophore. The author proposed a model to explain a novel function for the signa in this species.
I have some concerns and comments:
1) Figures did not label properly, for example, Figure 1, (B) was not typed! In figure 2, 3 and 5, more labels should be added to mark the specific components of the structure!
2) The (B) in figure 1 was partially duplicated from (A) in Figure 1 that was published by the author last year (Xochipiltecatl D, Baixeras J, Cordero CR. 2021. Atypical functioning of female genitalia explains monandry in a butterfly. PeerJ 9:e12499 https://doi.org/10.7717/peerj.12499), at least, this should be mentioned in the text!
3) The (D) in figure 2, was probably just a high magnification of (E) of figure 4 that was also published last year (mentioned above)!
4) In page 6, L191-193, “…..these interactions between the signum and the spermatophore occur before or very soon after the end of copulation.” The spermatophore was deposited in the bursa copulatrix during copulation, not before copulation, why here it used “before” or I have a misunderstanding here!
5) I prefer the author to discuss more concerning the evolutionary meaning of the signum structure in L. aripa in the discussion section.
Author Response
Dear Editor:
We thank reviewer 3 for their suggestions and criticism that helped us to improve our manuscript.
Here we provide a point-by-point response to their commentaries:
“1) Figures did not label properly, for example, Figure 1, (B) was not typed! In figure 2, 3 and 5, more labels should be added to mark the specific components of the structure!”
We have added the missing label to figure 1, as well as labels to critical structures in figures 2, 3 and 5. We have modified the captions of these figures to take into account the changes.
“2) The (B) in figure 1 was partially duplicated from (A) in Figure 1 that was published by the author last year (Xochipiltecatl D, Baixeras J, Cordero CR. 2021. Atypical functioning of female genitalia explains monandry in a butterfly. PeerJ 9:e12499 https://doi.org/10.7717/peerj.12499), at least, this should be mentioned in the text!”
It is right that figure 1B corresponds to the same corpus bursae of figure 1A in our PeerJ paper. However, strictly speaking, since we had multiple photographs of some specimens, the photograph included in our manuscript is not the same (we can provide the original photograph of the PeerJ paper if necessary), although it is a “zoom-in” of the same corpus bursae taken from the same angle. Although we mention in the first paragraph of Materials and Methods that the original observations reported in our manuscript were “mostly obtained from the study of the same specimens employed in” the PeerJ paper, we understand the concern of the reviewer and have decided to explain the relation between figure 1B and the previously published photograph in the figure caption in the following way:
“Figure 1. Bursa copulatrix of the butterfly L. aripa. (A) Cleared female abdomen showing the location and area occupied by the corpus bursae (cb), ductus bursae (db), ductus seminalis (ds) and signum (s). (B) Isolated CB of a virgin female. Figure 1B is a different photograph of the same CB whose photograph was published as figure 1A in reference [2]. Scale bars: A = 1 mm, B = 200 µm.”
“3) The (D) in figure 2, was probably just a high magnification of (E) of figure 4 that was also published last year (mentioned above)!”
The reviewer is right, figure 2D is the same as 4E of the PeerJ paper and this was acknowledged in the figure caption and the Acknowledgements section of our manuscript. Strictly speaking, this is the only previously published photograph that we pretend to use in our paper. As we informed the editorial office of Insects, previous to submission, we asked the editorial office of PeerJ for permission and they mention that we just need to cite the original source. For your information, this is the response mail of PeerJ:
“Alicia Santos <support@peerj.com> lun, 25 abr, 23:16
|
|
Dear Carlos,
Thank you for your email. Your article, "Atypical functioning of female genitalia explains monandry in a butterfly", was published as an open access article distributed under the terms of the Creative Commons Attribution License, which permits unrestricted use, distribution, reproduction and adaptation in any medium and for any purpose provided that it is properly attributed. For attribution, the original author(s), title, publication source (PeerJ) and either DOI or URL of the article must be cited.
Regards,
Alicia Santos
PeerJ Support”
“4) In page 6, L191-193, “…..these interactions between the signum and the spermatophore occur before or very soon after the end of copulation.” The spermatophore was deposited in the bursa copulatrix during copulation, not before copulation, why here it used “before” or I have a misunderstanding here!”
We have modified this sentence and we hope that it’s meaning is clear now:
“The observation of the imprint of the signum on the spermatophore in females sacrificed a few minutes after finishing mating indicates that the interaction between the signum and the spermatophore occurs before the end of copulation (once the spermatophore is fully formed but before separation of the couple) or very soon after separation of the couple.”
5) I prefer the author to discuss more concerning the evolutionary meaning of the signum structure in L. aripa in the discussion section.
We thank the reviewer for their suggestion. We realize that this was a missing point in our discussion. We added a new paragraph (the last of the Discussion) and a reference to address this important point:
“Comparative evidence suggests that signa originally evolved to break spermatophore envelopes in polyandrous species, and that the evolution of monandry selects for a loss of signa [16]. However, in L. aripa the evolution of monandry is associated with the evolutionary loss of the muscular sheath of the CB [2]. The lack of these muscles not only prevents the breaking of spermatophores, but also the ability to exert a pressure on the spermatophore that could help the passage of sperm cells from the sperm sac to the spermatheca. This could explain why L. aripa females did not lose their signum and why the signum acquired a novel function that helps the transfer of sperm cells from the spermatophore to the spermatheca. It will be interesting to investigate if the signum of L. aripa has evolved modifications that improve its (proposed) function as a valve, by comparison with the signa of closely related polyandrous species that digest the spermatophore.”
In the revised version, we highlighted in yellow the deleted parts and in green the new parts.
We hope that you find the revised version of our manuscript acceptable for publication and look forward to hearing from you.
Best regards,
Carlos Cordero
Round 2
Reviewer 3 Report
No more comments on the revised version of the manuscript